# SagaScale: A Realistic, Scalable, and High-Quality Long-Context Benchmark Built from Full-Length Novels

## Abstract

Large Language Models (LLMs) have shown significant progress, but understanding long and complex documents remains challenging. Many long-context benchmarks have been proposed, but they face several limitations, including task realism, data scalability, and data quality. We introduce **SagaScale, a realistic, scalable, and high-quality long-context benchmark built from full-length novels**. The entire benchmark is constructed using an automated data collection pipeline that utilizes **external resources** (e.g., Wikipedia pages) to curate question-answer pairs. Critically, these external resources are provided only for benchmark construction and not during evaluation, which allows LLMs to curate complex questions that go beyond what they can answer during evaluation. SagaScale is also bilingual and offers the largest context length to date, with average token counts exceeding 250K for English novels and 320K for Chinese novels. Our evaluation across 12 frontier LLMs and three long-context methods — Naïve RAG, Agentic RAG, and Long Context — yields key insights, including: (1) Directly supplying the full context to the LLM can outperform other methods by a large margin; (2) Most LLMs still struggle with lengthy contexts, but Gemini-2.5-Pro stands out as an exception; and (3) Agentic RAG effectively addresses the retrieval bottleneck in Naïve RAG. Finally, we publicly release the SagaScale benchmark and our data collection codebase to facilitate future research.

## 1 Introduction

Large Language Models (LLMs) have achieved remarkable success across a variety of natural language processing (NLP) tasks (OpenAI et al., 2024; Grattafiori et al., 2024). A key remaining challenge, however, is long context understanding: the ability to process, reason over, and synthesize information from lengthy and complex documents. To guide progress in this area, the community has increasingly focused on creating dedicated long-context benchmarks (Zhang et al., 2024; Hsieh et al., 2024; Yuan et al., 2024; Bai et al., 2025; Yen et al., 2025).

However, existing long-context benchmarks have several limitations. For ease of construction and evaluation, many benchmarks rely on synthetic tasks (Kamradt, 2024; Hsieh et al., 2024). Such synthetic tasks, while useful for initial evaluation, often fail to capture the complexity of real-world scenarios. To improve realism and maintain quality, other benchmarks (e.g., Bai et al., 2025) rely on heavy human annotation. While this approach yields high-quality data, it is often prohibitively costly and time-consuming, limiting its scalability for adaptation to new domains or training set curation. To automate data annotation, recent efforts like LaRA (Li et al., 2025) attempt to generate QA pairs from isolated document chunks. However, this leads to simpler, locally focused questions and potential factual errors. Additionally, LaRA relies on manually tuned seed examples and prompts for each task type, which also limits its scalability and question diversity.[1]

Therefore, existing long-context benchmarks (Kamradt, 2024; Kuratov et al., 2024; Wang et al., 2025; Bai et al., 2025; Li et al., 2025) face a fundamental conflict among task realism, data scalability, and data quality. To this end, we introduce **SagaScale, a realistic, scalable, and high-quality long-context benchmark built from full-length novels**.

---

[1] For example, over 75% of questions in LaRA's 128K book reasoning task start with "why".

| Benchmark | Realistic Task | Data Scalability | Multilingual | Context Length | Evaluation Task |
|---|---|---|---|---|---|
| RULER (Hsieh et al., 2024) | ✗ | ✓ | ✗ | — | Synthetic |
| Counting-Stars (Song et al., 2025) | ✗ | ✓ | ✓ | — | Synthetic |
| Loogle (Li et al., 2024a) | ✓ | ✗ | ✓ | <32K | Multitask |
| Longbench-v2 (Bai et al., 2025) | ✓ | ✗ | ✗ | <128K | Multitask |
| LaRA (Li et al., 2025) | ✓ | ◖ | ✗ | <128K | Single-doc QA |
| NovelQA (Wang et al., 2025) | ✓ | ✗ | ✗ | <200K | Fiction QA |
| **SagaScale (Ours)** | ✓ | ✓ | ✓ | 228K | Fiction QA |

Table 1: Comparison of SagaScale with existing benchmarks. Context lengths represent median token counts. The half-filled circle (◖) denotes limited data scalability, where humans must manually tune seed examples and prompts for each task type even when using LLMs for direct data generation (Li et al., 2025).

To achieve scalability without sacrificing quality, we introduce an automated, novel data collection pipeline that leverages LLMs to generate and filter QA pairs. In the generation phase, we provide the LLM with both the novel's text and **external resources** (e.g., Wikipedia articles) for broader context. Crucially, such asymmetry allows the LLM to generate questions that are much more complex than those it can answer during evaluation, where access is restricted to the novel's text alone. After generation, we employ a multi-stage filtering process that discards factually incorrect, unrealistic, or contaminated QA pairs (Balloccu et al., 2024), where external knowledge sources (e.g., Google Search) still play key roles. By systematically validating our benchmark against six quality criteria, we demonstrate that our data collection pipeline consistently produces high-quality QA pairs.

To the best of our knowledge, SagaScale is the first long-context benchmark that combines high data quality with full realism and scalability. SagaScale is also bilingual and offers one of the largest context lengths to date, with average token counts exceeding 250K for English novels and 320K for Chinese novels[2] (Figure 5).

Finally, we assess three representative methods on SagaScale, each embodying a distinct strategy for handling long contexts: (1) Long Context, where a long-context language model directly processes the entire document and answers the question in a single pass (Liu et al., 2025); (2) Naïve Retrieval-Augmented Generation (RAG), which retrieves a fixed set of text chunks with the highest embedding similarity to the query, then generating an answer based on these chunks (Zhao et al., 2022; Gao et al., 2024); and (3) Agentic RAG, an extension of Naïve RAG that empowers an agent to iteratively refine retrieval across multiple rounds before producing the final answer (Singh et al., 2025; Liang et al., 2025; Jin et al., 2025). We summarize our core contributions as follows:

- **Dataset**: We present a novel, automated data collection pipeline that leverages external resources to generate QA pairs, providing a foundation for at-scale evaluation and training set construction.

- **Benchmark**: We introduce SagaScale, a realistic, scalable, and high-quality bilingual long-context benchmark featuring one of the largest context lengths to date.

- **Analysis**: We evaluate three representative long-context approaches using a wide range of LLMs, and provide an in-depth analysis of their performance, yielding key insights.

## 2   RELATED WORK

**Long-context Approaches.** Methods for addressing long contexts broadly fall into two categories: (1) direct long-context processing, where a long-context language model (LCLM) processes the entire input; and (2) workflow-based approaches, which employ external workflows to manage long contexts (Liu et al., 2025). In the first category, significant progress has been recently made in

---

[2]All token counts in this paper are calculated using the tokenizer from `https://huggingface.co/deepseek-ai/DeepSeek-R1-0528`, unless stated otherwise.

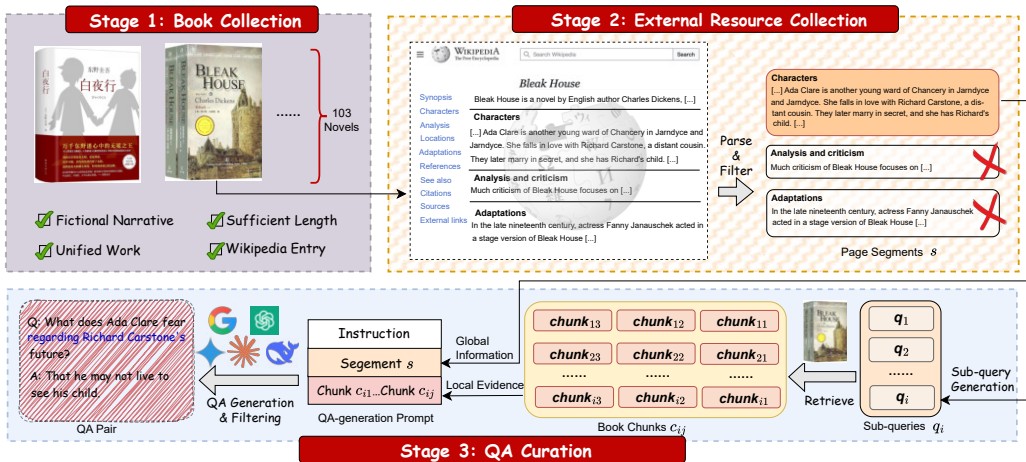

Figure 1: Data collection pipeline of SagaScale.

extending the context length of LLMs, with models now supporting context lengths up to 128K tokens (OpenAI et al., 2024a; DeepSeek-AI et al., 2025b), 200K tokens (Anthropic, 2025; AI et al., 2025), and even 1M tokens (Comanici et al., 2025; GLM et al., 2024). In the second category, retrieval-augmented generation (RAG) has been widely adopted (Gao et al., 2024; Fan et al., 2024). Based on the number of retrieval rounds in the workflow, RAG methods can be further divided into two groups: (1) single-round RAG (Ma et al., 2023; Wang et al., 2023; Shi et al., 2024), where a fixed set of text chunks is retrieved and used to generate the answer; and (2) iterative RAG (Singh et al., 2025; Jiang et al., 2025; Jin et al., 2025), where the model perform multiple rounds of retrieval to refine its retrieved data before generating the final answer. In this paper, the three evaluated methods correspond to: (1) direct long-context processing (Long Context); (2) single-round RAG (Naïve RAG); and (3) iterative RAG (Agentic RAG).

**Long-context Benchmarks.** Long-context benchmarks can be divided into two categories: (1) synthetic benchmarks, which are artificially constructed to isolate and measure specific long-context capabilities; and (2) real-world benchmarks, which are derived from naturally occurring data and reflect practical applications (Liu et al., 2025). While synthetic benchmarks are widely used for initial evaluation (Tay et al., 2021; Kamradt, 2024; Levy et al., 2024; Hsieh et al., 2024; Kuratov et al., 2024; Song et al., 2025; Ling et al., 2025), they often fail to predict downstream performance (Yen et al., 2025). To bridge this gap, many real-world long-context benchmarks have been developed (An et al., 2024; Wang et al., 2025; Bai et al., 2025), but they still face several limitations. First, the threshold for "long context" has increased rapidly due to advancements in LCLMs, making some existing benchmarks inadequate for evaluating current models (Kočiský et al., 2018; An et al., 2024; Bai et al., 2024). Second, real-world benchmarks typically rely on heavy human annotation (Karpinska et al., 2024; Wang et al., 2025; Zhang et al., 2024; Bai et al., 2025), which is unscalable due to prohibitively high costs and time consumption. While recent efforts like LaRA (Li et al., 2025) have attempted to automate data annotation, their scalability and data quality (e.g., diversity) remain limited. We summarize these limitations and highlight our benchmark's strengths in Table 1.

## 3 DATA

### 3.1 DATA COLLECTION

Figure 1 provides an overview of our data collection pipeline for SagaScale, which consists of three stages: Book Collection, External Resource Collection, and QA Curation.

**Stage 1: Book Collection.** We collect 103 full-length novels, comprising 77 in English and 26 in Chinese. The collection includes public domain books primarily sourced from Project Gutenberg[3]

---

[3]https://www.gutenberg.org/

and copyrighted books obtained from the web. Each book is selected based on four criteria: (1) the book must be a fictional narrative to reduce contamination risk, (2) it must be sufficiently long, typically exceeding 100K tokens, (3) it must be a single, unified work rather than an anthology of loosely related stories, and (4) it must have an informative Wikipedia (or Baidu Baike) entry. Together, the collected books feature complex storylines and substantial lengths (Table 9), enhancing the benchmark's overall difficulty and realism.

**Stage 2: External Resource Collection.** Existing long-context benchmarks are built primarily from the source documents themselves, while neglecting valuable *external resources* such as community-written summaries. These resources distill key information, provide deep insights, and broaden the overall context—all while being significantly shorter than the original documents. These characteristics make them ideal candidates for crafting high-quality QA pairs.

To collect external resources, we first pair each novel with its Wikipedia (or Baidu Baike) page. Then, we parse each page to extract self-contained segments (e.g., the sections titled "Synopsis"). To ensure questions derived from these segments are answerable using only the book text, we employ an LLM to discard any segment that contains extrinsic information (e.g., adaptations) or subjective interpretations (e.g., criticism).

**Stage 3: QA Curation.** To obtain high-quality QA pairs, we decompose QA curation into two sub-stages: generation and filtering. In generation, we design a specialized RAG pipeline to ground external resources to the book text, and then use rigorous prompts to generate QA pairs. After generation, we apply three distinct filters to enforce correctness, realism, and non-contamination.

**Stage 3.1: QA Generation.** For each page segment $s$ obtained from Stage 2, we start with retrieving relevant fixed-length chunks from the book. However, due to the lengthy and complex nature of the page segments, the original segment $s$ cannot serve directly as a query. To address this, we adopt a multi-query generation approach (Kostric & Balog, 2024; Li et al., 2024b). First, an LLM generates concise sub-queries $q = \{q_1, q_2, \ldots, q_n\}$ from $s$, each targeting a distinct aspect of $s$. Then, we use each $q_i$ to retrieve the three most relevant book chunks $\{\text{chunks}_{i1}, \text{chunks}_{i2}, \text{chunks}_{i3}\}$. Importantly, this approach enhances retrieval efficiency and diversity, while also adaptively scaling the total number of retrieved chunks based on the complexity of the page segment.

After the retrieval step, we concatenate each chunk list $\{\text{chunks}_{i1}, \text{chunks}_{i2}, \text{chunks}_{i3}\}$ with the original page segment $s$. This forms a comprehensive context that includes two layers of granularity: the broader, global information from the page segment ($s$) and the specific, local evidence from the book chunks ($\text{chunks}_{ij}$). We pass this context to DeepSeek-R1 (DeepSeek-AI et al., 2025a) to generate QA pairs (refer to Appendix A.4.1 for prompts).

**Stage 3.2: QA Filtering.** Our QA filtering process consists of three steps: Correctness Verification, Realism Assurance, and Contamination Filtering.

**Correctness Verification.** Since QAs are generated by LLMs using only page segments and book chunks, errors can arise from model mistakes, flawed segments, or insufficient context. To address this, we use GPT-4o Search Preview (OpenAI, 2025) with web search capability to verify each QA pair. Given each QA pair and the name of its corresponding novel, the LLM searches online, attempts to find relevant information, and returns a binary judgment (correct/incorrect) or states no relevant information is found. Adopting a conservative approach, we retain only QA pairs with confirmed correct answers.

Importantly, this filtering step does not make the benchmark inherently easier. Verifying a QA pair is fundamentally simpler than solving one, since the model is given the correct answer for validation. The model can also benefit from unrestricted web access, a knowledge source far exceeding that of the source books and Wikipedia.

**Realism Assurance.** During the QA generation sub-stage, the model is unaware of the real-world popularity of the questions it generates, which can result in unnatural or rare questions. Consequently, an additional filtering step is required to ensure the realism of our benchmark.

To estimate a question's popularity, we use an LLM to extract its keywords and then perform Google search with these keywords. The popularity score is defined as the total number of search results, which serves as an effective proxy for real-world user interests. Then, to account for differences in online presence across novels, we apply filtering on a per-book basis: for each novel, we keep

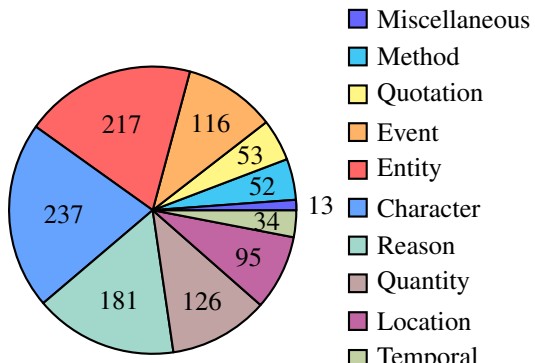

Figure 2: Distribution of questions across question types.

| Range | <128K | 128K–256K | 256K–512K | 512K–1M | Total |
|---|---|---|---|---|---|
| QA Pairs | 165 | 386 | 252 | 321 | 1124 |
| Novels | 20 | 41 | 21 | 21 | 103 |

Table 2: Distribution of QA pairs and novels across different token count ranges.

only questions with a score at least 0.01% of the highest-scoring question for that book. Refer to Appendix A.2.1 for a case study of this step.

**Contamination Filtering.** The novels, along with their related online resources, are often seen by LLMs during training (Zhu et al., 2015; Rae et al., 2020; Gao et al., 2020). This poses a risk of data contamination, where models may answer questions by solely recalling memorized knowledge rather than demonstrating true long-context understanding.

To mitigate this issue, a straightforward filtering step is implemented. We select four state-of-the-art LLMs with extensive world knowledge: GPT-4.1 (OpenAI, 2025), DeepSeek-R1 (DeepSeek-AI et al., 2025a), Claude-Opus-4 (Anthropic, 2025), and Gemini-2.5-Pro (Comanici et al., 2025). Then, we prompt each LLM to answer all questions without access to the books. If any of these models can answer a question correctly, we remove that question from the benchmark.

## 3.2 DATA VERIFICATION

SagaScale is built using the data collection pipeline described above (see A.2.2 for examples). To ensure benchmark quality, we validate it against six key criteria: Diversity (questions cover a wide range of question types), Non-contamination (models struggle to answer correctly using only parametric knowledge), Difficulty (questions are challenging), Answerability (questions are solvable using only the book text), Objectivity (Questions are objective and have verifiable answers), and Correctness (no factual errors in questions or answers).

- **Diversity:** Following Kočiský et al. (2018), we classify questions into types based on narrative theory. As shown in Figure 2, the questions are well-distributed across these types, indicating a balanced and varied set of questions. For details on question types and the full classification procedure, refer to Appendix A.3.

- **Non-contamination:** We evaluate LLMs in a closed-book setting (i.e., without access to the books) to test for data contamination, and find that the highest accuracy achieved is only 11.7% (Table 4). This demonstrates the effectiveness of our contamination filtering step and confirms that high performance requires genuine comprehension rather than pure memorization (Balloccu et al., 2024).

- **Difficulty:** We confirm the benchmark's challenging nature in Section 4 through experiments. While state-of-the-art models demonstrate strong capabilities, their performance remains well below perfect accuracy, leaving a substantial margin for improvement.

- **Answerability & Objectivity & Correctness:** We randomly sample 100 instances and ask our authors to manually verify them against the remaining criteria. All sampled questions are found to be answerable and objective, and 95 out of 100 instances are fully correct.

### 3.3 STATISTICS

**QA Curation Statistics.** Through our rigorous data collection pipeline, we ultimately collect 1,124 high-quality QA pairs, containing 613 for English novels and 511 for Chinese novels. Initially, the QA generation step produces 11,584 QA pairs (7,760 in English and 3,824 in Chinese), most of which are filtered out during contamination filtering.

**Token Count Distribution.** We categorize all QA pairs and novels into four length ranges based on book token counts: less than 128K, 128K–256K, 256K–512K, and 512K–1M. As shown in Table 2, both QA pairs and novels are distributed relatively evenly across these ranges. For detailed statistics for each novel, refer to Table 9.

## 4 EVALUATION

### 4.1 SETUP

**Methods.** We assess three representative methods on SagaScale: Long Context, Naïve Retrieval-Augmented Generation (RAG), and Agentic RAG. Below, we provide a detailed description of each method. Refer to Appendix A.4.2 for the specific prompts.

- **Long Context (LC).** For each question $q_i$, we create the model input by concatenating the instruction, the corresponding book text $t_i$, and the question $q_i$. When the input fits within the context window, the model generates an answer. Otherwise, we directly record a failure without any truncation attempts.
- **Naïve RAG (NR).** We first index each book by segmenting $t_i$ into 512-token chunks and embedding them with mE5-large (Wang et al., 2024). For each question $q_i$, we retrieve the top 3 chunks with the highest embedding similarity to $q_i$. Using these chunks as the context, we prompt each model to either generate an answer or flag the question as unanswerable.
- **Agentic RAG (AR).** This method builds upon Naïve RAG by employing an agentic workflow (Singh et al., 2025; Liang et al., 2025; Jin et al., 2025). Starting from the question $q_i$, the model repeatedly issues search queries to retrieve text chunks from the book. Once it deems the gathered information sufficient, a final answer is generated. We cap the process at a maximum of 8 retrieval requests. For a fair comparison, we use the same retrieval setup as in Naïve RAG.

**Models.** We evaluate 12 powerful LLMs, including open-source models like Qwen3-235B-A22B (Yang et al., 2025), DeepSeek-V3 (DeepSeek-AI et al., 2025b), DeepSeek-R1 (DeepSeek-AI et al., 2025a), and proprietary models such as GPT-4o (OpenAI et al., 2024a), GPT-4.1 (OpenAI, 2025), o1 (OpenAI et al., 2024b), o4-mini (OpenAI, 2025), Claude-4 (Anthropic, 2025), and Gemini-2.5 (Comanici et al., 2025). All open-source models are deployed natively without context window extension (e.g., no YaRN (Peng et al., 2024)), while the proprietary models are accessed via their respective APIs. We set the default decoding temperature to 0.0 for all models.

**LLM-as-a-Judge.** Following Yen et al. (2025), we employ GPT-4o (OpenAI et al., 2024a) to evaluate model-generated answers. To confirm its reliability, we manually annotate 100 samples produced during evaluation[4] and compare our annotations with GPT-4o's judgements. We find a Cohen's Kappa of 0.92, indicating strong agreement.

### 4.2 RESULTS & ANALYSIS

Table 3 presents the main evaluation results. Below, we analyze the results from multiple perspectives, with particular focus on Long Context (LC) and Agentic RAG (AR).

---

[4]All QA pairs sampled for LLM-as-a-Judge verification are confirmed to be completely correct.

| Model | Max Len | Naïve RAG | Agentic RAG | Long Context |
|---|---|---|---|---|
| Qwen3-235B-A22B[†] | 32K | 44.8 | 50.4 | 0.0[§] |
| Qwen3-235B-A22B[‡] | 32K | 46.0 | 46.3 | 0.0[§] |
| DeepSeek-V3-0324[†] | 128K | 49.4 | 52.5 | 9.3[§] |
| DeepSeek-R1-0528[‡] | 128K | 48.8 | 40.3 | 7.8[§] |
| GPT-4o-2024-11-20[†] | 128K | 45.8 | 53.3 | 7.6[§] |
| o1-2024-12-17[‡] | 200K | 51.4 | 61.1 | 14.5[§] |
| o4-mini-2025-04-16[‡] | 200K | **52.3** | 60.4 | 13.9[§] |
| Claude-Sonnet-4-20250514[†] | 200K | 47.6 | 61.9 | 10.8[§] |
| Claude-Opus-4-20250514[†] | 200K | 47.2 | 63.5 | 13.0[§] |
| Claude-Sonnet-4-20250514[‡] | 200K | 48.3 | 63.1 | 10.3[§] |
| Claude-Opus-4-20250514[‡] | 200K | 47.6 | **64.8** | 13.9[§] |
| GPT-4.1-2025-04-14[†] | 1M | 50.7 | 58.1 | 72.1 |
| GPT-4.1-mini-2025-04-14[†] | 1M | 48.8 | 50.1 | 49.6 |
| Gemini-2.5-Flash-2025-06-17[†] | 1M | 43.5 | 52.9 | 68.0 |
| Gemini-2.5-Flash-2025-06-17[‡] | 1M | 47.1 | 53.6 | 72.5 |
| Gemini-2.5-Pro-2025-06-17[‡] | 1M | 50.1 | 60.6 | **79.6** |
| **Average** | — | 48.1 | **55.8** | 27.7 |

[†] Non-reasoning models.

[‡] Reasoning models.

[§] Some problems exceed the model's maximum context length. These processing failures are included in the reported score.

Table 3: Model performance (%) across three tasks: Naïve RAG (NR), Agentic RAG (AR), and Long Context (LC). The highest score for each task is highlighted in **bold**, and the second highest is underlined.

**Overall comparison of methods.** As shown in Table 3, Agentic RAG (AR) outperforms Naïve RAG (NR) across nearly all models, demonstrating the advantage of agentic retrieval over static, single-pass retrieval. When comparing Long Context (LC) with NR and AR, we observe no advantage for LC due to context window limits. However, when restricted to problems that fit within each model's context window (Table 5), LC surpasses NR and AR by 10.9% and 4.4% on average, respectively. The superiority of LC is most pronounced for the top-performing 1M-context models (GPT-4.1 and Gemini-2.5 Series), where LC performance consistently surpasses even the highest AR score (64.8%). Therefore, we conclude that the simple approach of directly supplying the entire context as model input can easily outperform multi-step retrieval-based workflows, provided that the LLM can effectively process contexts of such length.

**All methods show declining performance as context length increases, with AR exhibiting the smallest drop.** Figure 3a plots the cumulative accuracy of each method at different book length thresholds. For all three methods, accuracy consistently declines as the context length increases. To further quantify this effect, we compute the gap in cumulative accuracy between problems up to 128K context and all those up to 1M context. The results reveal a 6.8% drop for NR, a 5.5% drop for AR, and a 6.3% drop for LC. Notably, AR exhibits the smallest relative decline, suggesting that iterative, agentic retrieval partially mitigates the impact of noise caused by extensive distracting content. Nevertheless, the magnitude of the decline across all methods remains considerable, highlighting the need for more robust LLMs and retrievers.

**Most models fail to maintain performance as context length increases, but Gemini-2.5-Pro succeeds.** We analyze LC performance for each 1M-context model, and plot their accuracy at various book length thresholds in Figure 3b. While a clear downward trend is observed for most models (GPT-4.1, GPT-4.1-mini, and Gemini-2.5-Flash), Gemini-2.5-Pro maintains stable accuracy across all lengths, standing out as an exception. This suggests that Gemini-2.5-Pro possesses unique capabilities for effectively processing extremely long contexts, which warrants further investigation.

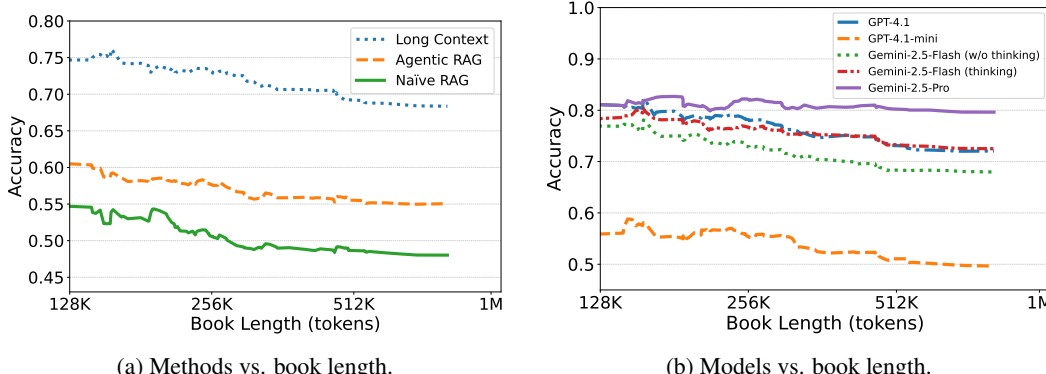

(a) Methods vs. book length.  (b) Models vs. book length.

Figure 3: Accuracy versus book length. **Left (a)**: A comparison of accuracy decay for the three evaluated methods. **Right (b)**: A breakdown of the Long Context performance for each model.

**Some models fail to adhere to formatting instructions in long contexts.** While LLMs are generally proficient at following simple formatting instructions (e.g., enclosing the answer in \boxed{}; see Appendix A.4.2) in short contexts (Ouyang et al., 2022), this ability can degrade drastically as context length increases. As shown in Table 6, a stark divergence is observed: top performers like Gemini-2.5 and GPT-4.1 maintain near-perfect formatting adherence, whereas models such as o1 and Claude-Sonnet-4 frequently fail. Although improved prompt engineering might mitigate this, the failure itself points to a critical weakness in certain models: formatting requires neither complex reasoning nor aggregation of information, but merely the recall of a simple instruction.

**Agentic RAG addresses the retrieval bottleneck in Naïve RAG.** In Naïve RAG (NR), model performance is largely limited by the quality of a fixed set of retrieved chunks. Poor retrieval prevents even strong LLMs from answering correctly, forming a *retrieval bottleneck*. This claim is supported by two observations: first, all models perform comparably in NR, with scores clustered between 43.5% and 52.3%; second, outcomes are highly polarized, with 66.7% of instances resulting in either 0% or 100% accuracy across all models (Table 8). Conversely, Agentic RAG (AR) exhibits no such clustering or polarization: AR scores vary widely from 40.3% to 64.8%, and only 25.4% of instances yield 0% or 100% accuracy. Furthermore, we find that 72.6% of the problems entirely unsolvable by NR are solved by at least one model using AR. These findings show that AR overcomes the retrieval bottleneck present in NR, demonstrating the potential of agentic retrieval to solve complex long-context problems.

**Low calibration error [5] is crucial for Agentic RAG performance, but unnecessary and even detrimental for Naïve RAG performance.** To investigate the relationship between calibration error rate (measured from NR) and model accuracy, we compute Spearman's rank correlation between two sets of variables:

- NR calibration error rates (Table 7), sorted in ascending order.
- Model accuracies for NR or AR (Table 3), sorted in descending order.

Interestingly, we find that better calibration in NR is *negatively* correlated with NR performance ($\rho_{NR}^{cal} = -0.5$) but *positively* correlated with AR performance ($\rho_{AR}^{cal} = 0.6$). While seemingly paradoxical, this finding aligns with our theoretical analysis (Appendix A.1). Intuitively, models prone to overconfidence (i.e., high calibration error rate) are penalized in AR because they tend to prematurely terminate the agentic retrieval process, whereas in NR, the same trait is beneficial because it can lead to occasional lucky guesses. Therefore, while low calibration error rate is unnecessary and even detrimental for the accuracy on common single-pass tasks (e.g., NR), it proves crucial for building strong search agents. We believe such distinction should be carefully considered when developing future LLMs.

---

[5]Following Wei et al. (2025), we define calibration error rate as the proportion of incorrect answers restricted to instances where the model attempts an answer. Questions where the model signals inability to solve are excluded.

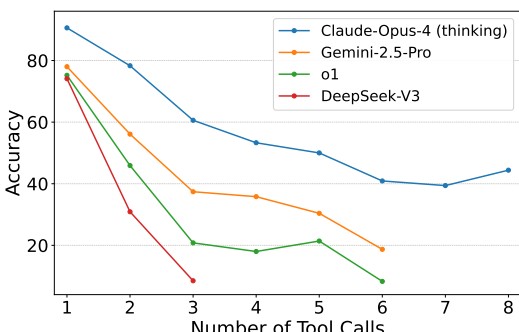

Figure 4: Model accuracy as a function of tool-call chain length in Agentic RAG. All models show a decline in accuracy when more tool calls are invoked. For statistical validity, data points representing fewer than 10 samples are omitted.

**Accuracy of Agentic RAG declines with longer tool-call chains.** We evaluate the AR accuracy of each model on problems where exactly $k$ tool calls are invoked, for $k \in \{1, 2, \ldots, 8\}$. As shown in Figure 4, when the number of tool calls increases, a drop in accuracy is observed across all models, including both top-performing models like Claude-Opus-4 and weaker ones like DeepSeek-V3. We hypothesize that this decline stems from two factors: problems requiring more tool calls are inherently more difficult, and the larger volume of retrieved chunks increases the risk of the model being misled by irrelevant or confusing information. This finding underscores the need for more robust agents and advanced workflows to maintain reliability over extended tool-call chains.

## 5 Conclusion

This paper introduces SagaScale, a bilingual long-context benchmark built from full-length novels. It addresses the fundamental conflict among realism, scalability, and quality, which existing benchmarks fail to resolve. At its core is a novel, automated data collection pipeline that generates complex QA pairs with **external resources** and applies filtering to ensure correctness, realism, and non-contamination. Our evaluation, conducted on 3 representative approaches across 12 frontier LLMs, offers a wealth of insights into their respective strengths and weaknesses. By releasing SagaScale and its full data pipeline to the community, we aim to set a new standard for long-context evaluation and accelerate the development of next-generation AI systems.

## 6 Limitations & Discussion

**Limited Dataset Size.** SagaScale consists of 1,124 QA pairs—a size sufficient for benchmarking but insufficient for training. This limitation stems from two main bottlenecks. First, our data annotation relies on manually collected novels and their Wikipedia (or Baidu Baike) links. While sophisticated web crawling could help, it is beyond the scope of this work. Second, our filtering process removes the vast majority of generated QA pairs to ensure non-contamination. While crucial for a fair comparison of frontier LLMs' long-context capabilities, a single round of contamination filtering with one model may suffice for training data construction. Future work could investigate simplifying our filtering process for training purposes.

**Limited Task Scope.** SagaScale is confined to a single task: question answering on novels. While this is an ideal real-world task for evaluating long-context understanding, expanding to a wider variety of tasks could be beneficial. We note that our core methodology, which incorporates external knowledge sources to generate test data, is not limited to novels — It can also be adapted for other domains, such as understanding large code repositories or even long videos (e.g., movies). Future work could explore these directions.

**Limited Language Coverage.** SagaScale is currently bilingual, including only novels in English and Chinese. Given the rich availability of novels and encyclopedic knowledge across a wide range of languages, future work could extend SagaScale to include more languages.

## Reproducibility Statement

To ensure reproducibility, we have publicly released all our code and data. The code repository offers detailed instructions on collecting data and replicating experiments. The released dataset contains the public-domain novels used in our benchmark, the generated question-answer pairs, and all evaluation verdicts (e.g., correct, incorrect) for each model and task. We also provide the evaluation setup and prompts in Section 4.1 and Appendix A.4.2, respectively.

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

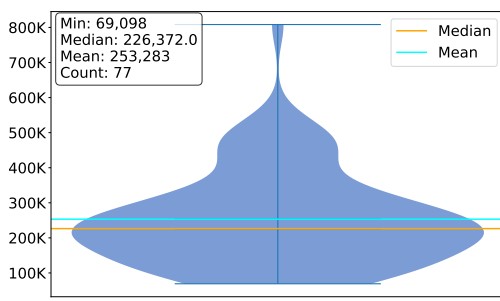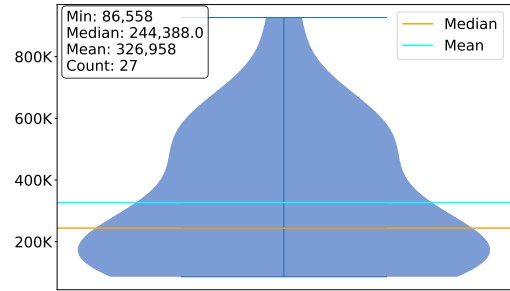

Figure 5: Violin plots illustrating the distribution of book token counts within the English (left) and Chinese (right) subsets of SagaScale.

# A APPENDIX

## A.1 THEORETICAL ANALYSIS

Below, we provide a theoretical analysis of how calibration error relates to the accuracy of Naïve RAG (NR) and Agentic RAG (AR).

Denote $a^{\mathrm{NR}}$ and $e^{\mathrm{NR}}$ as the probabilities of producing a correct or incorrect answer in NR, respectively. Note that $a^{\mathrm{NR}}+e^{\mathrm{NR}}$ may be less than 1, since the model can flag a question as unanswerable. Similarly, let $a_i^{\mathrm{AR}}$ and $e_i^{\mathrm{AR}}$ be the probabilities of producing correct or incorrect answer in AR after exactly $i$ retrieval requests. Here, $a_i^{\mathrm{AR}} + e_i^{\mathrm{AR}}$ may also be less than 1, as the model can continue to issue more retrievals instead of answering.

Throughout this analysis, we assume that the outcomes of each retrieval step are independent; that is, the probability of any outcome at step $i$ does not depend on the outcomes of previous steps. Therefore, the accuracy of NR is simply $a^{\mathrm{NR}}$, and the accuracy of AR is given by

$$\sum_{i=0}^{N} \left( \prod_{j=0}^{i-1} (1 - a_j^{\mathrm{AR}} - e_j^{\mathrm{AR}}) \right) a_i^{\mathrm{AR}}$$

where $N$ denotes the maximum number of retrievals allowed.

If accuracy alone is considered, NR does not penalize overconfidence, as increasing $e^{\mathrm{NR}}$ alone does not affect the NR accuracy which always remains $a^{\mathrm{NR}}$. In fact, overconfidence can even be implicitly rewarded in NR, as higher $e^{\mathrm{NR}}$ may result in increased $a^{\mathrm{NR}}$ due to lucky guesses. Conversely, AR is highly sensitive to overconfidence: large values of $e_i^{\mathrm{AR}}$ can substantially decrease AR accuracy, since the weighting coefficients for $a_i^{\mathrm{AR}}$ decrease exponentially as $i$ increases.

To illustrate this, consider a simplified case where $a^{\mathrm{NR}} = a$, $a_i^{\mathrm{AR}} = a$ for all $i \geq 0$, and $e^{\mathrm{NR}} = e$, $e_i^{\mathrm{AR}} = e$ for all $i \geq 0$. Let $N \rightarrow \infty$, then the accuracy of NR is simply $a$, while the accuracy of AR becomes the sum of a geometric series:

$$\sum_{i=0}^{\infty} (1 - a - e)^i a = \frac{a}{a + e}$$

## A.2 Case Study

### A.2.1 Case Study of Realism Assurance

To illustrate the effectiveness of our realism assurance step, we present several examples in Figure 6. The questions marked in red are representative of those discarded during the realism assurance step: they either delves into negligible details peripheral to the main narrative (e.g., "direction must travelers take"), or are structured more like an exam quiz than a natural user inquiry (e.g., "In which city ..., the same location where ..."). In contrast, the questions marked in green are more concise, natural, and focus on important characters (e.g., "Dr. Holmes", "Sara Macallan") or plot points (e.g., the defense's failure to prove intent). As a result, they achieve higher popularity scores, indicating a stronger alignment with genuine user interest.

---

**Retained and filtered question examples in realism assurance**

Q: What direction must travelers take to use the longer but easier route between Burgdale and Shadowy Vale? (The Roots of the Mountains)

Q: In which city was Gaston de Lancy buried, the same location where his mother resided and was falsely claimed to be ill? (The Trail of the Serpent)

Q: How many children does Dr. Holmes have? (Mrs. Dalloway)

Q: Why was the defense unable to prove Sara Macallan intended to obtain arsenic? (The Law and the Lady)

---

Figure 6: Examples of questions retained (in green) or filtered out (in red) during the realism assurance step. The retained questions are more likely to be asked by real users, while the filtered ones are often too rare or unnatural.

### A.2.2 Case Study of SagaScale

We show representative QA examples from SagaScale in Figure 7, including one in English and one in Chinese.

---

**QA examples in SagaScale**

Q: After Toad confesses his identity and actions to the Engine Driver, what specific event occurs that leads the driver to allow Toad to escape?
A: passing through a tunnel
(Oliver Twist)

Q: 云天明赠送给程心的恒星DX3906在哪个纪元被发现拥有行星?
A: 威慑纪元61年
(三体系列)

---

Figure 7: Examples of questions retained (in green) or filtered out (in red) during the realism assurance step. The retained questions are more likely to be asked by real users, while the filtered ones are often too rare or unnatural.

A.3 QUESTION CLASSIFICATION

Given the large size of our benchmark, we employ GPT-4o to automatically classify questions. The prompt below provides all category names, definitions, and illustrative examples to guide the model.

```
Prompt for Question Classification

## Task
Given a question, classify it into one of the following categories.

Put the category name in a single \boxed{} block (e.g., \boxed{Location}).

## Category Definitions

Location: Questions targeting places.
Character: Questions targeting roles in the narrative.
Reason: Questions targeting the reasons or motivations.
Method: Questions targeting how something was accomplished.
Event: Questions targeting specific occurrences or plot points.
Entity: Questions targeting specific entities (not characters or locations).
Quantity: Questions targeting amounts or numbers.
Temporal: Questions targeting when an event occurs or its duration.
Quotation: Questions targeting specific phrases or terms in the text.
Miscellaneous: Questions that do not fit into the categories above.

## Examples

Location: Where did Oliver express his grief after being punished for defending his
mother?
Character: Who does Valeria privately suspect of intentionally poisoning Sara
Macallan with arsenic?
Reason: Why did the jury return a 'not proven' verdict instead of 'not guilty' for
Eustace Macallan?
Method: How did Abel Magwitch escape from the prison ship?
Event: What happened to the heiress George Vavasor was engaged to before her money
could benefit him?
Entity: What item of clothing hides Miserrimus Dexter's physical deformity during
Valeria's visit?
Quantity: How many children does John Boucher have?
Temporal: How long did Orlando serve as ambassador to Constantinople before being
elevated to a Duke?
Quotation: What phrase described Oliver's trial period as an apprentice with Mr.
Sowerberry?
Miscellaneous: What is the nickname of the child adopted by Joseph Peters?

## Input
{question}
```

Figure 8: Prompt template used for question classification. All category names, definitions, and examples are available in the prompt.

A.4 PROMPTS

A.4.1 DATA COLLECTION PROMPTS

Below, we present the prompt template used for each stage of our data collection process. The QA generation step is particularly important, so we provide a detailed explanation of the key rules specified in the prompt (Figure 11):

- *The question should focus only on fictional elements without involving any real-life information.* This rule, combined with the segment filtering step in Stage 2, prevents

questions that require outside information, thereby enforcing answerability. Thus, the collected external resources are used only to provide book-based knowledge rather than all generic facts.

- *The question should be highly objective and specific, leading to a unique answer*. This ensures question objectivity and answer verifiability.

- *The question must be multi-hop, requiring in-depth understanding and detailed knowledge. It should span multiple passages or different parts of the same passage to be answered correctly.*. This promotes questions that require holistic understanding, detailed knowledge, and complex reasoning, thereby enhancing the difficulty of our benchmark.

- *The answer must be a single number, entity or short phrase that directly addresses the question.* This rule reinforces objectivity and simplifies evaluation. It is also crucial for answer correctness, as it discourages lengthy or complex answers that may introduce inaccuracies.

---

**Prompt for filtering article segments**

```
## Task
Given a Wikipedia text that contains information about the novel {novel_name},
determine if it is a summary (e.g., plot summary, character summary) of the novel.
Additional Requirements:
- The text must be highly objective. Excessive subjective contents, e.g., opinions,
speculation, interpretative analysis, subjective commentary, should fail the test.
- The text must focus on the novel's fictional elements. Excessive inclusion of
real-world topics, e.g., literary study, historical context, author biography,
publication history, cultural impact, awards, adaptations, academic analysis, should
fail the test.

Output \boxed{YES} or \boxed{NO} after careful examination.

## Input
{article_segment}
```

Figure 9: Prompt template used for filtering segments parsed from Wikipedia (or Baidu Baike) articles. The model is instructed to identify segments that are objective summaries of the novel's fictional elements, ensuring relevance and answerability.

---

**Prompt for multi-query generation**

```
## Task
Given a Wikipedia text for the novel {novel_name}. You should break down the text
into fine-grained, specific parts and write a concise summary for every part. Each
summary should contain one to two sentences.

Your output should be in {novel_language} and adhere to the following format for each
summary: "<summary>
your_summary
</summary>".

## Input
{article_segment}
```

Figure 10: Prompt template used for multi-query generation. The model is instructed to break down the lengthy article segment into concise summaries, each serving as a query to retrieve relevant book chunks for QA generation (Stage 3.1).

```
Prompt for generating QA pairs

## Task
Given several passages for the novel {novel_name}. The first passage is from
Wikipedia, while the rest are novel snippets.

Generate a single question-answer pair. Requirements:
1. The question should focus only on fictional elements of the novel (e.g.,
characters, plots) without involving any real-life information (e.g., author,
reviews, writing styles, other novels).
2. The question should be highly objective and specific, leading to a unique answer.
However, any unnecessary hint or context should be excluded.
3. The question must be **multi-hop**, requiring in-depth understanding and detailed
knowledge. It should span multiple passages or different parts of the same passage to
be answered correctly.
4. The answer must be a single number, entity or short phrase that directly addresses
the question.
5. Both the question and the answer should be unambiguous when the passages are not
provided.
6. If some previously generated question-answer pairs are provided, the new pair
should be completely different from those pairs (by focusing on distinct characters,
plots, aspects, etc.).

If such pair is available, put the generated question and answer in two separate
\boxed{} blocks. Both of them should be in {novel_language}.

If no such pair is available, your output should be a single \boxed{END} block. Do
not think too long.

## Input
{passages}
```

Figure 11: Prompt template used for generating QA pairs. The model is instructed to create QA pairs that are answerable (i.e., solvable using only the book text), objective, difficult, and diverse.

```
Prompt for verifying QA pairs

## Task
Given a QA pair for the novel {novel_name}, you should check if it is correct.
Requirements:
- The **answer** must be accurate and must not omit any core information.
- The **question** must not contain any factual errors or misleading content.

Output your decision as \boxed{PASS} or \boxed{FAIL}.

If no relevant information is found on the web, output \boxed{FAIL}.

## Input
{question}

{answer}
```

Figure 12: Prompt template used for verifying the factual accuracy of QA pairs. The model is instructed to ensure the correctness of the QA pair through web search.

---

**Prompt for extracting keywords**

```
## Task
Given a question for the novel {novel_name}, you should identify all keywords in the
question. Requirements:
- All unnecessary words that do not affect the core intent should be removed.
- Each keyword should be kept in its shortest form.

Your output should be several \boxed{} blocks, each containing exactly one keyword.

## Input
{question}
```

---

Figure 13: Prompt template used for extracting keywords from a question. The extracted keywords are used to perform a Google search to estimate question popularity.

---

**Prompt for Contamination Filtering**

```
## Task
Given a question for the novel {novel_name}, you should answer the question based on
your knowledge of the novel.

Your output should be in {novel_language} and adhere to the following format for the
final answer: "<answer>
{final_answer_text}
</answer>".

## Input
{question}
```

---

Figure 14: Prompt template used for filtering contaminated QA pairs. We provide the model with the novel name and the question, and ask it to craft an answer based solely on its internal knowledge. Note that this prompt is also used in closed-book evaluation.

### A.4.2 EVALUATION PROMPTS

Below, we present the prompt template for LLM-as-a-Judge and each of the three evaluation settings.

```
Prompt for LLM-as-a-Judge

## Task
Given a question for the novel {novel_name}, the corresponding ground truth answer,
and a model-generated answer, you should judge whether the model-generated answer is
correct.

The model-generated answer is correct if and only if:
- It directly addresses the question.
- No part of it contradicts the ground truth answer.
- It captures the essence or key points in the ground truth answer (note that a
lexical exact match is not required, but the semantic meaning should be preserved).

Output your decision as either \boxed{CORRECT} or \boxed{INCORRECT}.

## Input
Question: {question}

Ground truth answer: {ground_truth_answer}

Model-generated answer: {model_generated_answer}
```

Figure 15: Prompt template used for judging the correctness of a model-generated answer against a ground truth answer.

```
Prompt for Long Context Evaluation

## Task
Given a question for the novel {novel_name}, you should answer the question based
solely on the full text of the novel.

Your output should be in {novel_language} and adhere to the following format for the
final answer: "<answer>
final_answer_text
</answer>".

## Book Text
{book_text}

## Question
{question}
```

Figure 16: Prompt template used for Long Context evaluation. We provide the model with the novel name, the full text of the book, and the question, and ask it to craft an answer based solely on the book text.

```
## Task
Given a question for the novel {novel_name}, you should answer the question based
solely on the provided contexts. Each context is a direct excerpt from the book, with
its location indicated by the source index ('idx').

If the question is answerable based solely on the provided contexts, you should
provide a final answer. Your output should be in {novel_language} and adhere to the
following format for the final answer: "<answer>
{final_answer_text}
</answer>".

Otherwise, output "<answer>
NONE
</answer>".

## Contexts
{contexts}

## Question
{question}
```

Figure 17: Prompt template used for Naïve RAG evaluation. We provide the model with the novel name, contexts retrieved from the book, and the question. The model is instructed to craft an answer based solely on the provided contexts, or respond with "NONE" if the information is insufficient.

```
## Task
Given a question for the novel {novel_name}, you should answer the question based
solely on retrieving contexts from the novel. To do this, you will serve as an agent
that iteratively request a search system to retrieve contexts, refine your query
based on the retrieved contexts, and ultimately generate the final answer when you
have sufficient information.

Specifically, every time a query is sent, you will receive a list of contexts, where
each context is a direct excerpt from the book with its location indicated by the
source index ('idx'). To return these contexts, the search system works by simply
retrieving the top-3 book chunks with the highest cosine similarities to the query,
so they might be noisy.

In each turn, you must provide either a single query or a single final answer. Your
output should be in {novel_language} and adhere to the following formats:
- For queries, use "<query>
query_text
</query>".
- For the final answer, use "<answer>
final_answer_text
</answer>".

Now, begin with the first turn.

## Question
{question}
```

Figure 18: Prompt template used for Agentic RAG evaluation. We provide the model with the novel name and the question, and instruct it to iteratively retrieve contexts and refine its query until it can provide a final answer.

| Model | Accuracy |
|---|---|
| Qwen3-235B-A22B[†] | 6.5 |
| Qwen3-235B-A22B[‡] | 5.5 |
| DeepSeek-V3-0324[†] | 8.2 |
| DeepSeek-R1-0528[‡] | 7.3 |
| GPT-4o-2024-11-20[†] | 5.2 |
| o1-2024-12-17[‡] | **11.7** |
| o4-mini-2025-04-16[‡] | 6.9 |
| Claude-Sonnet-4-20250514[†] | 6.2 |
| Claude-Opus-4-20250514[†] | 4.0 |
| Claude-Sonnet-4-20250514[‡] | 6.4 |
| Claude-Opus-4-20250514[‡] | 10.0 |
| GPT-4.1-2025-04-14[†] | 4.4 |
| GPT-4.1-mini-2025-04-14[†] | 3.8 |
| Gemini-2.5-Flash-2025-06-17[†] | 5.8 |
| Gemini-2.5-Flash-2025-06-17[‡] | 5.2 |
| Gemini-2.5-Pro-2025-06-17[‡] | 10.4 |

[†] Non-reasoning models.

[‡] Reasoning models.

Table 4: Model performance (%) in the closed-book setting, where the model has no access to the books and must answer questions based on its internal knowledge.

| Model | Max Len | NR | AR | LC |
|---|---|---|---|---|
| Qwen3-235B-A22B[†] | 32K | — | — | — |
| Qwen3-235B-A22B[‡] | 32K | — | — | — |
| DeepSeek-V3-0324[†] | 128K | 55.8 | 53.3 | **63.6** |
| DeepSeek-R1-0528[‡] | 128K | **57.0** | 46.7 | 53.3 |
| GPT-4o-2024-11-20[†] | 128K | 48.8 | 53.5 | **65.9** |
| o1-2024-12-17[‡] | 200K | 55.6 | **64.4** | 51.7 |
| o4-mini-2025-04-16[‡] | 200K | 57.1 | **64.1** | 49.5 |
| Claude-Sonnet-4-20250514[†] | 200K | 53.8 | **63.1** | 53.8 |
| Claude-Opus-4-20250514[†] | 200K | 51.1 | **65.8** | 64.9 |
| Claude-Sonnet-4-20250514[‡] | 200K | 51.4 | **61.4** | 55.2 |
| Claude-Opus-4-20250514[‡] | 200K | 49.5 | 64.8 | **74.3** |
| GPT-4.1-2025-04-14[†] | 1M | 50.7 | 58.1 | **72.1** |
| GPT-4.1-mini-2025-04-14[†] | 1M | 48.8 | **50.1** | 49.6 |
| Gemini-2.5-Flash-2025-06-17[†] | 1M | 43.5 | 52.9 | **68.0** |
| Gemini-2.5-Flash-2025-06-17[‡] | 1M | 47.1 | 53.6 | **72.5** |
| Gemini-2.5-Pro-2025-06-17[‡] | 1M | 50.1 | 60.6 | **79.6** |
| **Average** | — | 51.5 | 58.0 | **62.4** |

[†] Non-reasoning models.

[‡] Reasoning models.

Table 5: Model performance (%) across three tasks: Naïve RAG (NR), Agentic RAG (AR), and Long Context (LC). Each model is evaluated on a unique subset of the benchmark which only includes problems that fit within its context window. *Consequently, scores are not directly comparable between different models.* The best-performing task for each model is highlighted in **bold**.

| Model | Max Len | Err Rate↓ |
|---|---|---|
| Qwen3-235B-A22B[†] | 32K | — |
| Qwen3-235B-A22B[‡] | 32K | — |
| DeepSeek-V3-0324[†] | 128K | 0.6 |
| DeepSeek-R1-0528[‡] | 128K | 13.9 |
| GPT-4o-2024-11-20[†] | 128K | 2.3 |
| o1-2024-12-17[‡] | 200K | 31.1 |
| o4-mini-2025-04-16[‡] | 200K | 15.2 |
| Claude-Sonnet-4-20250514[†] | 200K | 26.7 |
| Claude-Opus-4-20250514[†] | 200K | 12.4 |
| Claude-Sonnet-4-20250514[‡] | 200K | 26.2 |
| Claude-Opus-4-20250514[‡] | 200K | 1.4 |
| GPT-4.1-2025-04-14[†] | 1M | 2.1 |
| GPT-4.1-mini-2025-04-14[†] | 1M | 2.2 |
| Gemini-2.5-Flash-2025-06-17[†] | 1M | **0.0** |
| Gemini-2.5-Flash-2025-06-17[‡] | 1M | 0.3 |
| Gemini-2.5-Pro-2025-06-17[‡] | 1M | 0.3 |

[†] Non-reasoning models.

[‡] Reasoning models.

Table 6: Parsing error rates in the Long Context (LC) task, where only problems that fit within each model's context window are considered. *Lower* values indicate stronger format adherence, and the lowest value is highlighted in **bold**.

| Model | NR↓ | AR↓ |
|---|---|---|
| Qwen3-235B-A22B[†] | 16.3 | 49.3 |
| Qwen3-235B-A22B[‡] | 26.0 | 53.7 |
| DeepSeek-V3-0324[†] | 19.0 | 47.5 |
| DeepSeek-R1-0528[‡] | 22.0 | 59.7 |
| GPT-4o-2024-11-20[†] | 17.6 | 46.7 |
| o1-2024-12-17[‡] | 18.3 | 38.7 |
| o4-mini-2025-04-16[‡] | 24.8 | 34.1 |
| Claude-Sonnet-4-20250514[†] | **11.7** | 35.6 |
| Claude-Opus-4-20250514[†] | 14.5 | 33.1 |
| Claude-Sonnet-4-20250514[‡] | 15.7 | 33.9 |
| Claude-Opus-4-20250514[‡] | 12.3 | **28.1** |
| GPT-4.1-2025-04-14[†] | 24.3 | 41.8 |
| GPT-4.1-mini-2025-04-14[†] | 39.7 | 49.8 |
| Gemini-2.5-Flash-2025-06-17[†] | 12.0 | 46.9 |
| Gemini-2.5-Flash-2025-06-17[‡] | 20.0 | 43.8 |
| Gemini-2.5-Pro-2025-06-17[‡] | 17.8 | 38.9 |

[†] Non-reasoning models.

[‡] Reasoning models.

Table 7: Calibration error rates for Naïve RAG (NR) and Agentic RAG (AR) tasks. In NR, a problem is treated as unanswered if the model flags it as unanswerable (see Section 4.1), whereas in AR, it is considered unanswered if more than the maximum of 8 search queries are issued. The lowest calibration error rate for each task is highlighted in **bold**.

| Methods | Polarization |
|---|---|
| Naïve RAG (NR) | 66.7 |
| Agentic RAG (AR) | 25.4 |
| Long Context (LC) | 34.6 |

Table 8: Proportion (%) of polarized outcomes (0% or 100% accuracy across all models) in different methods. Among the three methods, Naïve RAG (NR) exhibits the highest polarization, while Agentic RAG (AR) and Long Context (LC) show significantly reduced polarization.

| Title | Language | Tokens | #QAs |
|---|---|---|---|
| *A Portrait of the Artist as a Young Man* | English | 113,002 | 12 |
| *A Tale of Two Cities* | English | 187,847 | 4 |
| *Across the Zodiac: The Story of a Wrecked Record* | English | 216,635 | 1 |
| *Adam Bede* | English | 297,822 | 11 |
| *Anna Karenina* | English | 489,802 | 10 |
| *Anne of Green Gables* | English | 143,117 | 12 |
| *Babbitt* | English | 180,959 | 1 |
| *Barnaby Rudge: A Tale of the Riots of Eighty* | English | 352,812 | 13 |
| *Bleak House* | English | 491,790 | 9 |
| *Brave New World* | English | 87,214 | 3 |
| *Brideshead Revisited* | English | 145,520 | 19 |
| *Can You Forgive Her?* | English | 419,072 | 8 |
| *Crime and Punishment* | English | 288,969 | 3 |
| *David Copperfield* | English | 497,757 | 14 |
| *Dombey and Son* | English | 498,783 | 8 |
| *Dracula* | English | 213,967 | 7 |
| *Emma* | English | 215,375 | 2 |
| *Gone with the Wind* | English | 544,052 | 6 |
| *Great Expectations* | English | 253,785 | 6 |
| *Jane Eyre: An Autobiography* | English | 257,206 | 1 |
| *Les Misérables* | English | 808,272 | 9 |
| *Little Dorrit* | English | 466,180 | 15 |
| *Main Street* | English | 236,938 | 5 |
| *Man and Wife* | English | 314,721 | 8 |
| *Mansfield Park* | English | 210,685 | 6 |
| *Marcella* | English | 339,614 | 6 |
| *Mary Barton: A Tale of Manchester Life* | English | 219,279 | 9 |
| *Middlemarch* | English | 434,792 | 11 |
| *Moby-Dick; or, The Whale* | English | 309,020 | 3 |
| *Mrs. Dalloway* | English | 88,704 | 11 |
| *Murder Must Advertise* | English | 153,295 | 2 |
| *New Grub Street* | English | 251,977 | 10 |
| *Nicholas Nickleby* | English | 463,064 | 5 |
| *North and South* | English | 248,470 | 13 |
| *Nostromo* | English | 233,197 | 15 |
| *Oliver Twist* | English | 226,372 | 2 |
| *Orlando: A Biography* | English | 106,267 | 2 |
| *Paul Clifford* | English | 252,075 | 1 |
| *Pride and Prejudice* | English | 170,016 | 2 |
| *Quo Vadis: A Narrative of the Time of Nero* | English | 287,683 | 10 |
| *Rebecca of Sunnybrook Farm* | English | 100,226 | 6 |
| *Robinson Crusoe* | English | 142,413 | 8 |
| *Shirley* | English | 299,751 | 8 |
| *Sons and Lovers* | English | 228,329 | 2 |
| *Tess of the d'Urbervilles* | English | 206,879 | 6 |
| *The Adventures of Huckleberry Finn* | English | 158,133 | 6 |
| *The Adventures of Tom Sawyer* | English | 101,736 | 2 |
| *The Angel of the Revolution: A Tale of the Coming Terror* | English | 188,538 | 8 |
| *The Gilded Age: A Tale of Today* | English | 213,972 | 3 |

| Title | Language | Tokens | #QAs |
|---|---|---|---|
| *The History of Tom Jones, a Foundling* | English | 469,959 | 15 |
| *The Invisible Man: A Grotesque Romance* | English | 69,098 | 8 |
| *The Last Man* | English | 239,038 | 19 |
| *The Law and the Lady* | English | 188,599 | 20 |
| *The Long Goodbye* | English | 151,143 | 8 |
| *The Moon and Sixpence* | English | 100,210 | 5 |
| *The Moonstone: A Romance* | English | 265,674 | 19 |
| *The Mysteries of Udolpho: A Romance* | English | 403,637 | 21 |
| *The Old Wives' Tale* | English | 299,707 | 4 |
| *The People of the Mist* | English | 193,417 | 1 |
| *The Picture of Dorian Gray* | English | 106,865 | 5 |
| *The Razor's Edge* | English | 148,535 | 2 |
| *The Red and the Black: A Chronicle of 1830* | English | 257,076 | 2 |
| *The Roots of the Mountains* | English | 212,977 | 5 |
| *The Scarlet Letter* | English | 117,841 | 6 |
| *The Sun Also Rises* | English | 96,280 | 1 |
| *The Trail of the Serpent* | English | 205,726 | 14 |
| *The Well at the World's End* | English | 304,222 | 13 |
| *The Wind in the Willows* | English | 83,485 | 10 |
| *The Woman in White* | English | 326,156 | 12 |
| *To the Lighthouse* | English | 93,387 | 2 |
| *Toilers of the Sea* | English | 198,712 | 11 |
| *Twenty Thousand Leagues Under the Seas* | English | 205,206 | 5 |
| *Twenty Years After* | English | 349,814 | 8 |
| *Uncle Tom's Cabin* | English | 263,615 | 4 |
| *Vanity Fair* | English | 424,757 | 10 |
| *Villette* | English | 274,495 | 21 |
| *White Fang* | English | 97,089 | 18 |
| 三个火枪手 | Chinese | 275,541 | 11 |
| 三体 | Chinese | 551,978 | 55 |
| 书剑恩仇录 | Chinese | 373,300 | 12 |
| 亮剑 | Chinese | 236,235 | 19 |
| 倚天屠龙记 | Chinese | 696,632 | 6 |
| 四世同堂 | Chinese | 480,143 | 9 |
| 围城 | Chinese | 156,627 | 46 |
| 基督山伯爵 | Chinese | 536,808 | 4 |
| 孽子 | Chinese | 93,595 | 4 |
| 孽海花 | Chinese | 191,706 | 18 |
| 官场现形记 | Chinese | 472,021 | 45 |
| 尼罗河上的惨案 | Chinese | 89,222 | 20 |
| 平凡的世界 | Chinese | 538,468 | 24 |
| 沙丘 | Chinese | 244,388 | 2 |
| 球状闪电 | Chinese | 105,150 | 15 |
| 白夜行 | Chinese | 190,339 | 25 |
| 白鹿原 | Chinese | 332,728 | 41 |
| 百年孤独 | Chinese | 155,996 | 3 |
| 解忧杂货店 | Chinese | 98,577 | 21 |
| 许三观卖血记 | Chinese | 86,558 | 8 |
| 达芬奇密码 | Chinese | 166,332 | 8 |
| 连城诀 | Chinese | 164,682 | 30 |
| 金瓶梅 | Chinese | 598,581 | 20 |
| 金粉世家 | Chinese | 648,839 | 17 |
| 飞狐外传 | Chinese | 322,264 | 42 |
| 骆驼祥子 | Chinese | 94,597 | 6 |

Table 9: List of novels in SagaScale. Token counts are calculated using the tokenizer from `https://huggingface.co/deepseek-ai/DeepSeek-R1-0528`.

