# OpenReview forum: "SagaScale: A Realistic, Scalable, and High-Quality Long-Context Benchmark Built from Full-Length Novels"
_ICLR.cc/2026/Conference — Submitted to ICLR 2026_

### Official Review · Reviewer_ULnD · 2025-10-24

**Soundness:** 3
**Presentation:** 2
**Contribution:** 3
**Rating:** 4
**Confidence:** 4

**Summary:**

This paper proposes a long-context evaluation framework, SagaScale, which is build from full-length novels. Its data creation pipeline collects more that 100 novels and corresponding external resources. Unlike existing benchmarks with real data, it employs an automatic QA generation and filtering, ensuring high-quality QA pairs. The experimental results demonstrate that their datasets are challenging for even recent LCLMs.

**Strengths:**

- This paper introduces an automated method for creating complex QA pairs by using external resources, e.g., Wikipedia during generation but not during evaluation. This information asymmetry effectively resolves the trade-off between realism, scalability, and quality.
- The benchmark employs a multi-stage filtering process to ensure correctness, realism, and non-contamination. This makes the evaluation more reliable.
- The authors tested three baselines, naive RAG, agentic RAG, and long context, with 12 LLMs on SagaScale.

**Weaknesses:**

- The reliance on a web-searching GPT-4o to verify QA pairs may not guarantee factual correctness within the novel’s context. The model might simply find corroborating text in an external resources, which doesn’t prove the answer is supported solely by the source text itself.
- The paper does not explain the relationship between the contamination filtering method and the closed-book evaluation results. It is unclear and I am wondering why models still achieve about 10% accuracy on the closed-book setting in Table 4 (see the first bullet point in the Questions section).
- The paper could be improved in its structure and clarity. For instance, their key insights are not well summarized in the Introduction section even though the primary focus of the paper is on benchmarking LLMs on their proposed datasets, leaving the reader to piece them together from later sections. I strongly recommend to explicitly highlight the most interesting findings in the section.

**Questions:**

Why do models obtain around 10% accuracy with the closed-book setting (Table 4) even though the Contamination Filtering in Section 3.1 removes questions that can be answered by any of four frontier models? If I understand correctly, both the filtering and the closed-book setting have the same procedure. Was this caused by model randomness? If so, I believe “non-contamination” is misleading.

---

### Official Review · Reviewer_KKMA · 2025-10-25

**Soundness:** 1
**Presentation:** 2
**Contribution:** 1
**Rating:** 2
**Confidence:** 5

**Summary:**

SagaScale introduces a bilingual long-context benchmark constructed from full-length novels, featuring 1,124 question-answer pairs across 103 novels (77 English, 26 Chinese) with context lengths averaging over 250K tokens for English and 320K tokens for Chinese. The authors propose an automated pipeline that leverages external resources like Wikipedia during question generation while restricting evaluation to novel text only, aiming to address what they perceive as limitations in existing benchmarks regarding task realism, data scalability, and quality. The paper evaluates 12 frontier LLMs using three long-context processing approaches: Long Context, Naïve RAG, and Agentic RAG.

**Strengths:**

**1. The benchmark provides genuinely ultra-long context evaluation.** With English novels averaging over 250K tokens and Chinese novels exceeding 320K tokens—some reaching 800K+ tokens—SagaScale represents one of the longest context benchmarks currently available. This is a meaningful contribution given the rapid advancement of long-context modeling capabilities in modern LLMs, and the bilingual coverage across English and Chinese adds valuable cross-lingual evaluation capacity that remains relatively uncommon in the field.

**2. The information asymmetry approach in data generation sounds interesting.** The use of external resources (Wikipedia) during question generation while withholding them during evaluation is conceptually interesting. This design enables the creation of more complex questions that potentially require genuine document understanding rather than relying on memorized knowledge, distinguishing it from straightforward extractive QA tasks.

**3. The experimental evaluation demonstrates reasonable comprehensiveness in scope.** The paper conducts evaluations across 12 frontier LLMs and compares three representative long-context processing methods, providing a relatively complete picture of current model capabilities under different processing paradigms. The systematic comparison of Long Context, Naïve RAG, and Agentic RAG approaches offers practical insights for practitioners.

**Weaknesses:**

**1. The literature review is severely inadequate, undermining the paper's fundamental motivation.** The authors claim existing benchmarks suffer from "insufficient task realism," "limited data scalability," and "data quality issues," but these assertions lack substantive support and mischaracterize the current state of the field. HELMET [1] already provides a comprehensive evaluation across seven diverse, application-centered categories, including multi-hop reasoning, temporal reasoning, and entity tracking—directly contradicting claims about synthetic tasks. ∞Bench [2] provides evaluation beyond 100K tokens across real-world tasks such as code debugging and mathematical reasoning. Loong [3] effectively balances realism, scalability, and quality through its multi-document QA framework, while LV-Eval [4] demonstrates that automated methods can indeed produce high-quality benchmarks at scale across five length levels up to 256K tokens. This insufficient engagement with existing work makes the claimed contributions appear overstated and the research motivation weak.

**2. The data quality assurance mechanisms contain fundamental flaws that compromise the reliability of benchmarks.** The paper's over-reliance on LLM generation followed by LLM verification—this "LLM-stacking" approach—cannot provide adequate quality guarantees. Only 100 samples (less than 9% of the total) were manually verified, which is statistically insufficient; a dataset of 1,124 QA pairs would require at least 300-400 samples. **More problematically, the paper explicitly states that quality inspectors are "our authors," seriously violating fundamental data annotation principles—standard practice requires independent third-party annotators to avoid confirmation bias and conflicts of interest**. No inter-annotator agreement metrics (Cohen's Kappa, Fleiss' Kappa, or Krippendorff's Alpha) are provided, and critical information about annotation procedures, costs, and standards is missing. While the paper reports Cohen's Kappa of 0.92 between GPT-4o as judge and human annotation, this is based on only 100 samples with unclear overlap with quality verification samples, and without baseline inter-human agreement for comparison. The circular dependency of using LLMs to both generate and verify QA pairs risks systematic bias going undetected if generation and verification models share similar error patterns.

**3. The claimed "scalability" advantage lacks substantive support and comes at quality costs.** True scalability should enable low-cost expansion to new domains or languages while maintaining quality. The contamination filtering step requires calling four top-tier commercial LLMs, which is hardly cost-effective, and manual collection of novels and Wikipedia links is still required, inherently limiting scalability. In contrast, LV-Eval [4] achieves genuine scalability while maintaining quality through techniques like Confusing Fact Insertion (CFI) and Keyword-Phrase Replacement (KPR). The paper's "scalability" primarily manifests as automation without the quality assurance mechanisms that would make it truly valuable.

**4. The experimental findings lack novelty and offer limited new insights.** The paper's main findings have been extensively documented in existing literature. That directly providing full context outperforms chunks has been confirmed in multiple studies, including ∞Bench, Loong, and LV-Eval, particularly when models have sufficient context windows. Gemini-2.5-Pro's excellence in long-context processing has been thoroughly validated in technical reports and multiple benchmarks. That Agentic RAG outperforms Naive RAG simply confirms consensus in the RAG field that iterative retrieval beats single-round retrieval, with extensive literature support and clear reasons (multi-turn RL training). The experiments provide another data point but lack profound insights that could advance community understanding.

**5. The task scope is overly narrow, severely limiting applicability and impact.** The paper acknowledges that SagaScale is limited to the single task of novel QA, which constrains the benchmark's generalizability and practical value. LongBench-v2 provides multitask evaluation across different long-context understanding scenarios; HELMET designs seven task categories for comprehensive assessment; Loong's multi-document framework applies to academic literature, legal cases, and financial reports. By focusing exclusively on novel comprehension, SagaScale's ability to become a widely adopted standard benchmark in the community is questionable, as real-world applications require diverse task types and domains.


**References:**

[1] Yen H, Gao T, Hou M, et al. Helmet: How to evaluate long-context language models effectively and thoroughly. Proceedings of ICLR, 2025.

[2] Zhang X, Chen Y, Hu S, et al. ∞Bench: Extending long context evaluation beyond 100k tokens. Proceedings of ACL, 2024.

[3] Wang M, Chen L, Cheng F, et al. Leave No Document Behind: Benchmarking Long-Context LLMs with Extended Multi-Doc QA. Proceedings of EMNLP, 2024.

[4] Yuan T, Ning X, Zhou D, et al. Lv-eval: A balanced long-context benchmark with 5 length levels up to 256k. arXiv preprint arXiv:2402.05136, 2024.

**Questions:**

**1. Could you provide complete annotation procedures and standards?** This should include annotator recruitment criteria and training procedures, detailed content of annotation guidelines, inter-annotator agreement metrics with at least two independent annotators, and annotation cost estimates including average time and cost per QA pair. These details are essential for assessing the benchmark's reliability and reproducibility.

**2. How were the 100 samples for manual verification selected?** Specifically, was stratified sampling used to ensure representativeness across different novels, question types, and difficulty levels? Random sampling from an imbalanced distribution may not adequately represent the full dataset's quality.

**3. What are the specific details of contamination filtering?** Please provide the actual performance of each of the four models in closed-book testing (individual model accuracy), clarify how "correct answer" is defined (exact match, fuzzy matching, semantic similarity with thresholds?), and explain why the stringent "any model correct" criterion was chosen over alternatives like "majority of models correct" that might better balance contamination control with selection bias.

**4. How are cases exceeding the model's context window handled in Long Context evaluation?** The paper mentions "directly recording failure," but is this a fair evaluation approach? Models with shorter context windows are inherently penalized regardless of their actual understanding capabilities. Were context compression, summarization techniques, or sliding window approaches considered as alternatives that might provide more meaningful comparisons across models with different architectural constraints?

---

### Official Review · Reviewer_tXpo · 2025-11-01

**Soundness:** 2
**Presentation:** 2
**Contribution:** 2
**Rating:** 4
**Confidence:** 4

**Summary:**

This paper introduces SagaScale, a novel long-context benchmark designed to address the limitations of existing ones. Built from full-length bilingual novels, it features unprecedented context lengths (avg. >250K tokens) and high-quality question-answer pairs generated via an automated pipeline. The paper's evaluation of 12 LLMs reveals that direct full-context processing is superior to RAG-based methods. Notably, Gemini-2.5-Pro excels where most models fail.

**Strengths:**

1. This benchmark includes unprecedented context lengths, bilingual support, and a scalable, automated pipeline for generating high-quality question-answer pairs.
2. Beyond just presenting a new dataset, the paper provides a rigorous evaluation of a wide range of state-of-the-art LLMs and three distinct long-context methods. This analysis establishes valuable baselines and offers crucial insights into the current landscape.

**Weaknesses:**

1. A key limitation is the narrow scope of the dataset, which consists solely of fictional novels. This domain is not representative of common long-context tasks in practice, which raises concerns about the practical relevance and generalizability of the evaluation.
2. During the QA generation phase, all question-answer pairs are generated by a single model, DeepSeek-R1. This could lead to a lack of diversity in the QAs and potentially create a bias that favors DeepSeek-R1's own evaluation. An ideal approach would be to have a list of models and randomly sample from them to generate the QAs.
3. The paper's approach to contamination filtering is a potential weakness. Given that many state-of-the-art LLMs are trained on extensive data that likely includes the source novels, this method may not be sufficient to fully eliminate contamination. A more rigorous approach would involve applying advanced decontamination techniques, like keyword substitution or paraphrasing, to the entire context rather than limiting the filtering to only the QA pairs.
5. The paper should justify the selection of GPT-4o for the evaluation. It is unclear why more capable models with larger context windows were not included, which could be a limitation of the comparative analysis.
6. The paper could be significantly strengthened by including an analysis of error types and the 'lost in the middle' phenomenon [1]. Such a qualitative analysis would offer deeper and more valuable insights than a discussion focused solely on correctness rates.
7. Missing citations for some long-context modeling benchmark works [2][3].

[1] Lost in the Middle: How Language Models Use Long Contexts. (TACL'23)

[2] BAMBOO: A Comprehensive Benchmark for Evaluating Long Text Modeling Capacities of Large Language Models. (COLING'24)

[3] Leave No Document Behind: Benchmarking Long-Context LLMs with Extended Multi-Doc QA. (EMNLP'24)

**Questions:**

please see Weaknesses

---

### Official Review · Reviewer_PExB · 2025-11-01

**Soundness:** 2
**Presentation:** 3
**Contribution:** 2
**Rating:** 4
**Confidence:** 4

**Summary:**

This paper introduces SagaScale, a realistic and scalable long-context QA benchmark constructed from full-length novels using an automated pipeline that generates high-quality, multi-step QA pairs. The benchmark evaluates different long-context methods (naïve RAG, agentic RAG, direct long context), providing comprehensive insights into their performance.

**Strengths:**

- The benchmark offers realistic, high-quality QA tasks.
- Presents a rigorous, multi-stage QA generation and filtering pipeline to ensure quality.
- Comprehensive evaluation including native RAG, agentic RAG, and long-context processing.
- Clear and well-structured presentation facilitates understanding.

These elements are crucial, as they significantly enhance the benchmark's realism and practical utility, enabling more effective evaluation of long-context language models, which are highly relevant to the community.

**Weaknesses:**

- Lack of transparency regarding the total dataset construction cost undermines one of the claimed benefits—cost efficiency compared to human annotation. There should be a more detailed comparison to the previous benchmark on the cost and scalability.
- Insufficient analysis of QA types: It remains unclear what capabilities models need to correctly answer questions—whether questions typically require retrieving information from a single segment, integrating multiple segments, or performing multi-step reasoning. Merely reporting general question types (e.g., character, event, location) does not adequately clarify the complexity or retrieval demands posed by each question. A deeper, more granular analysis would better illustrate the nature of these questions and the model capabilities required to answer them successfully.
- Missing detailed error analysis: The paper lacks quantitative analysis of error types (e.g., factual errors, ambiguities) to clearly support claims about the QA pairs' high quality. Detailed error categorization would strengthen validation.
- Dependence on LLM-as-a-Judge impacts reproducibility and accessibility.
- Related work discussion lacks clarity, particularly the differentiation between synthetic and real-world tasks. Additional relevant citations recommended:

[1] Liu et al., Lost in the middle: How language models use long contexts. 2023.

[2] Yen et al., HELMET: How to Evaluate Long-Context Language Models Effectively and Thoroughly. 2025.

[3] Li et al., Long-context LLMs Struggle with Long In-context Learning. 2024.

[4] Lee et al., Can Long-Context Language Models Subsume Retrieval, RAG, SQL, and More? 2024.

[5] Zou et al., On Many-Shot In-Context Learning for Long-Context Evaluation. 2025.

[6] Shaham et al., ZeroSCROLLS: A Zero-Shot Benchmark for Long Text Understanding. 2023.

[7] Wang et al., Ada-LEval: Evaluating long-context LLMs with length-adaptable benchmarks. 2024.

**Questions:**

- What is the precise cost of the dataset construction pipeline?
- Why record a failure without truncation attempts for longer contexts?
- How expensive is running the LLM-as-a-Judge for evaluations?
- Given only 9% retention of generated questions, how necessary are external resources versus randomly sampling novel chunks to generate questions?
- How many novels present potential copyright issues?

**Details Of Ethics Concerns:**

In the paper, the author mentions that some copyrighted novels are collected.

---

### Meta-Review · Area_Chair_kZQC · 2025-12-06

**Summary:**

The reviewers' concerns can be summarized as follows:
- Some claimed contributions lack substantive support
- Insufficient insightful experimental analysis
- The literature review is severely inadequate, undermining the paper's fundamental motivation

**Reviewer Concerns:**

The concerns are still outstanding as the authors do not provide rebuttal.

**Reviewer Scores:**

The reviewers would keep their scores due to lack of rebuttal.

---

### Decision · Program_Chairs · 2026-01-26

Reject